# Semi-Empirical Time-Dependent Parameter of Shear Strength for Traction Force between Deep-Sea Sediment and Tracked Miner

**DOI:** 10.3390/s22031119

**Published:** 2022-02-01

**Authors:** Wei Yi, Feng Xu

**Affiliations:** 1School of Mathematics and Physics, University of South China, Hengyang 421001, China; 2School of Civil Engineering, Central South University, Changsha 410083, China; yi.wei@csu.edu.cn

**Keywords:** time-dependent cohesion, traction force, deep-sea sediment, tracked miner, rheology

## Abstract

Based on our direct shear creep experiment and the direct shear rheological constitutive model, a semi-empirical time-dependent parameter of the shear strength is obtained by Mohr–Coulomb shear strength theory, and different time-dependent traction force calculations between deep-sea sediment and a tracked miner are conducted by the work-energy principle. The time-dependent traction force calculation under its influencing factors, including the time, track shoe number, and grounding pressure, are analyzed and proved to be valid by the traction force experiment of a single-track shoe. The results show that the time-dependent cohesion force obtained by a semi-empirical way can be easily used to deduce the time-dependent traction force models under the different grounding pressure distributions and applied into deep-sea engineering application conveniently; the verified traction force models by the traction force experiment of a single-track shoe illustrate that traction force under the decrement grounding pressure distribution is the worst among the four kinds of grounding pressure distributions and suggested for evaluating the most unfavorable traction force and calculating the trafficability and stability of the deep-sea tracked miner.

## 1. Introduction

Many countries shift their attention from the land to the deep sea, which is abundant with mineral resources, due to the gradual lack of non-renewable land resources with the development of social economy. In the deep sea, there are nearly 1500 billion tons of mineral resources, including poly-metallic nodules, cobalt-rich shells, and poly-metallic sulfides [1,2,3]. Therefore, these countries target ocean exploitation as their strategic development direction. At present, there are three types of commonly used deep-sea mining systems: drag bucket mining system, continuous rope bucket mining system, and mining system with a tracked miner and pipeline lifting device (i.e., hydraulic lifting pipeline mining system). The tracked miner is one of the most critical pieces of equipment of the hydraulic lifting pipeline mining system, adopted mainly by China because of its low cost and high mining efficiency [4,5,6]. The tracked miner encounters extremely special deep-sea environments, such as more obvious rheological performance of sediment than land soil [7] and complex deep-sea topography (e.g., gullies, ditches, and slopes). The special environments inevitably result in the change of grounding pressure distribution and the lack of traction force for the tracked miner, and the tracked miner eventually fails the normal walk. It will cause severe turnover of the tracked miner due to the deep sinkage and breakdown of the whole mining system [8]. Therefore, it is significant for the safety and stability of the deep-sea mining system to study time-dependent characteristics of deep-sea sediment and the traction force of the tracked miner under different grounding pressure.

Currently, the research on time-dependent characteristics of the deep-sea sediment is mainly carried out from two aspects, i.e., the creep experiment and the rheological constitutive model. The research on the traction force mainly includes three methods, i.e., the traction force theory, the traction experiment, and the related simulation. In terms of the time-dependent characteristics of the deep-sea sediment, a direct shear creep experiment was conducted for analyzing the deep-sea simulant sediment with Burgers creep model and obtaining related rheological parameters of the direct shear rheological model [9]. A tri-axial compression creep experiment was studied for getting the creep curves of the deep-sea simulant sediment under the same confining pressure and different vertical compression, and then the creep curves were employed to identify the parameters by the different rheological models for determining the most proper compression rheological model of deep-sea sediment [10]. Xu et al. [7] discovered the compression-shear coupling effect between compression creep displacement and shear creep displacement by the compression-shear coupling creep experiment, and then a compression-shear coupling rheological model was deduced and proved to be reliable. In regard to the traction force of the tracked vehicle, the track-terrain interaction theory is often adopted mostly involving the Bekker’s theory (for brittle soil) [11], Janosi–Hanamoto’s theory (for plastic soil) [12] and Wong’s theory (for both brittle and plastic soil) [13], and a little involvement of rheological constitutive model (for cohesive soil) [14]. For example, Zhao et al. [15] categorized the compression model of soil based on the soil theory and offroad vehicle-terrain theory and introduced an improved sinkage model of the brittle soil by analyzing the Bekker’s model and ultimate balance theory. Wu et al. [16] established a traction force model of a tracked miner on the deep-sea soft sediment by studying the cohesive action between a grouser and the sediment, which revealed the influence of parameters of the sediment and sizes of the tracked miner structure on the traction force. Wang et al. [17] tested the applicability of two kinds of empirical models of shear stress-displacement to the deep seabed and promoted a new empirical model for saturated and plastic soil; Xu et al. [18] initially employed a compression-shear coupling rheological model into the analyses of the sinkage and thrusting force of a tracked miner and deduced a new turning traction force. Li et al. [19] obtains a relationship between the grouser height and the water jet based on elastic-plastic traction force model aiming at the cause of sticky soil shaped on the track. Experimentally, Xu et al. [20] discussed the relationship between the slippage and traction force and determined the optimal grouser height by analyzing the motion of simplified track shoes and the track shoe experiment of different grouser heights. Shin et al. [21] discussed the loss mechanism of the lateral thrusting force by a traction force experiment with different shape ratios. Baek et al. [22] evaluated the traction performance by testing the slipping sinkage of the track shoes. Furthermore, in terms of the simulation of traction force, Rubinstein et al. [23] studied a transporting tracked vehicle with dynamics software LMS-DADS and established a multi-body dynamic simulation model for calculating the traction force on different locations of the track shoes. Yang et al. [24] studied the influence of the shearing ratio and grouser size on the traction force based on the simulation model of track shoe and soil. Li et al. [25] built a 3-D simulation model with McKyes–Ali software to analyze the interaction between interval track shoes and the soil and verify the interaction laws. Summarily, the existing study on the track force is mainly about the elastic-plastic and rheological performance of the deep-sea sediment, seldom involving the time-dependent performance of the shear strength in deducing the time-dependent traction force under different grounding pressure distributions, which brings the inconvenience of applying the temporal traction force model into the deep-sea mining engineering.

In this paper, the direct shear creep experiment is conducted for obtaining the semi-empirical time-dependent parameters in the shear strength based on the analyses of direct shear rheological model and Mohr–Coulomb shear strength theory. The time-dependent shear strength parameter will be employed into deducing the models of traction force based on the work-energy principle under different grounding pressure distributions (uniform, linear, and sine) and analyzing the influence of parameters such as the time, number of the track shoe, and grounding pressure distribution on the traction force. The research results could provide scientific basis for designing and optimizing the crawler as well as evaluating the trafficability of the tracked miner.

## 2. Time-Dependent Transformation of Shear Strength Parameter

### 2.1. Direct Shear Creep Experiment of Simulant Deep-Sea Sediment

The deep-sea soft sediment burdens a vertical pressure (e.g., grounding pressure) and a horizontal force (e.g., traction force) when the tracked miner walks. Therefore, it is necessary to study the direct shear creep effect of deep-sea soft sediment (provided that grounding pressure is not varying over time). The simulant sediment with the most similar physical and mechanical properties to deep-sea sediment (undisturbed sediment) is prepared to satisfy a large number of tests by mixing different betonies with water in various ratios. The main parameters measured by our research team are shown in Table 1 [9]. It can be seen that the main physical and mechanical parameters of simulant sediment and undisturbed sediment are close to each other and satisfied the requirement of the experiment [9].

The deep-sea simulant sediment is sampled into a standard cylinder shape of 60 mm × 30 mm (Figure 1) [10] by a ring knife, considering that its water content (*w* = 165.6%) is between the plastic limit and the liquid limit. The sheer creep test of simulant sediment (with constant compressive stress) is conducted on a self-developed pressure-shear creep test device (Figure 2) [20] by our research team. The simulant sediment sample is placed in a shear box and burdens the effect of vertical weights (compressive stress *σ*) and horizontal weights (shear stress *τ*), i.e., the pressure-shear loading. Since the average grounding pressure *σ*_0_ (*σ*_0_ = 5 kPa) is the minimum standard for determining compressive stress and the average shear strength (*τ_b_* = 6 kPa) is the maximum standard for determining shear stress, six different groups of constant compressive stress (*σ* = 5 kPa, 10 kPa, 15 kPa, 20 kPa, 25 kPa, 30 kPa) and six different groups of shear stress (*τ* = 1 kPa, 2 kPa, 3 kPa,4 kPa, 5 kPa, 6 kPa) are arranged and combined into 36 groups in total for direct shear creep experiment.

For every group, the shear creep curves in the horizontal direction are obtained from a displacement test system comprising a NS-WY02 high-precision displacement sensor, a signal amplifier, and a display (computer). Noticeably, NS-YB data acquisition software is adopted for data storage due to the shortcoming of real-time data display [7].

### 2.2. Direct Shear Rheological Constitutive Model

Figure 3a and Figure 4a show the typical shear creep curves of the simulant sediment under different constant compressive stresses *σ* (*σ* = 5 kPa [20] and 10 kPa) as examples, which are obtained under different constant shear stresses *τ*. Based on the characteristics analyses of the curves, Burgers rheological model, i.e., shear stress (*τ*)-displacement (*s*)-time (*t*) equation (Equation (1)) is adopted to fit these experimental creep curves (*s*-*t*) and the four rheological parameters (*K*_1_, *K*_2_, *β*_1_ and *β*_2_) in the model can be auto-fitted and determined by Sigma-plot software with fitted curved 3D surface consisting of experimental creep curves and are relevant with different constant compressive stresses (*σ*).
(1)$$s(\tau ,t)=\tau \left[\frac{1}{K_{1}}+\frac{t}{\beta_{1}}+\frac{1}{K_{2}}\left(1-e^{-tK_{2}/\beta_{2}}\right)\right]$$


As an example, Figure 3b and Figure 4b show the curved 3D surface in *τ*-*s*-*t* space fitted by Equation (1) for the specific constant *σ* (*σ* = 5 kPa [20] and 10 kPa), where solid points are experimental data. By resorting to “Dynamic Fit Wizard” in the Sigma-plot software and inputting Equation (1) with user-defined function, the fitted Burgers rheological model parameters can be obtained (such as for the *σ* = 5 kPa, *K*_1_ = 7.36 MPa, *K*_2_ = 1.82 MPa, *β*_1_ = 7380 MPa·s, *β*_2_ = 22,900 MPa·s, coefficient of determination R-Square is 0.987). Table 2 lists all of the fitted shear creep parameters under different *σ* as well as R-Square and illustrates that the fitted shear creep parameters are the functions of *σ* and increase with *σ*. Therefore, the shear rheological Equation (1) can express the relationships between shear stress *τ*, shear displacement *s*, compressive stress *σ*, and time *t*, i.e., *τ* = *τ*(*s*, *σ*, *t*) and can be rewritten as Equation (2), i.e., direct shear rheological constitutive model, where the expressions of *K*_1_(*σ*), *K*_2_(*σ*), *β*_1_(*σ*), and *β*_2_(*σ*) are obtained by fitting data in Table 2 [20] and described by Equation (3).
(2)$$s(\tau ,\sigma ,t)=\tau \left[\frac{1}{K_{1}(\sigma )}+\frac{t}{\beta_{1}(\sigma )}+\frac{1}{K_{2}(\sigma )}\left(1-e^{-tK_{2}(\sigma )/\beta_{2}(\sigma )}\right)\right]$$
(3)$$\left\{\begin{array}{l} K_{1}(\sigma )=0.002873\sigma^{3}-0.1312\sigma^{2}+2.373\sigma +3.363\\ K_{2}(\sigma )=0.02327\sigma^{2.14}+3.882\\ \beta_{1}(\sigma )=0.008\sigma^{3}+0.5629\sigma^{2}-4.483\sigma +47.89\\ \beta_{2}(\sigma )=0.0011\sigma^{2}-0.00672\sigma +0.0316 \end{array}\right.$$


### 2.3. Time-Dependent Parameters of Shear Strength

The shear stress (*τ*)-shear displacement (*s*) relationships (Equation (4)) are deduced by Janosi–Hanamoto based on Mohr–Coulomb shear strength theory and is often applied into deducing traction force of kinds of tracked vehicles [13]. For better deduction and analyses of traction force of different grounding pressure distributions under the deep-sea tracked miner, the direct shear rheological model is adopted to obtain the time-dependent parameters of Mohr–Coulomb shear strength and easily used to deduce traction force equation.
(4)$$\tau =\left[c+\sigma \tan\varphi \right](1-e^{-s/\kappa })$$
where *c* is cohesion force, *φ* is friction angle and κ is shearing deformation module of the deep-sea simulant sediment. Since *φ* and *κ* are too small (*φ* is 1.72° and *κ* is 0.00424 mm), the two parameters can be regarded to be constants and not relevant with time.

Generally, the parameters in Equation (4) are determined based on the Figure 5, where the black dots represent the peak shear resistance and the straight line is fitted to the black dots. Obviously, the coordinate (*σ*, *τ*) of the intersection point between the straight line and *τ*-axial is (0, *c*), i.e., *τ*(0) = *c*. In order to obtain a time-dependent cohesion force *c*, let the *σ* in the Equation (2) be zero and shear stress *τ* will be the function of the shear displacement *s* and time *t* without the compressive stress *σ*, i.e., *c*(*s*, *t*) = *τ*(*s*, 0, *t*). Since shear displacement *s* = *vt* and shear velocity *v* is constant, *c* is only the function of time *t*, i.e., *c* = *c*(*t*).

Hence, the parameters *K*_1_(*σ*), *K*_2_(*σ*), *β*_1_(*σ*) and *β*_2_(*σ*) become constant and the value of the parameters are as follows [20].
(5)$$\left\{\begin{array}{l} K_{1}(0)=3.363\\ K_{2}(0)=3.882\\ \beta_{1}(0)=47.89\\ \beta_{2}(0)=0.0316 \end{array}\right.$$


Accordingly, Equation (2) can be rewritten into Equation (6) given by
(6)$$s=vt=c(t)\cdot \left[0.297+\frac{t}{47.89}+\frac{1}{3.882}\left(1-e^{-122.85t}\right)\right]$$


By means of modifying Equation (6) into a function of time *t*, the time-dependent cohesion force *c* is given by
(7)$$c(t)=vt/\left[0.297+\frac{t}{47.89}+\frac{1}{3.882}\left(1-e^{-122.85t}\right)\right]$$


## 3. Time-Dependent Traction Force of Deep-Sea Miner Crawler

### 3.1. Traction Force Model

Figure 6 plots the simplified model of the miner’s motion on the deep-sea sediment [26]. It can be seen that the whole crawler comprises several hinged track shoes with uniformed distribution. To simplify calculation of the traction force, it is assumed that every track shoe *I* (*i* = 1, 2, …, *n*) is a “T” type with a grouser thrusting and shearing the sediment and moves at the same horizontal displacement. When the tracked miner moves forwards straightly at a constant velocity *v*, every vertical grouser of the track shoe *i* develops a horizontal shear displacement *s* (i.e., slippage *i*Δ). Choosing the whole crawler as an analysis object, as shown in Figure 7, the contacting force (i.e., grounding pressure *σ*(*x*)) on the track shoe is affected by the complex deep-sea topographies such as deep-sea mountains and ditches because of different contact conditions between track shoe surface and an idler. If every track shoe *i* moves under the action of traction force *T_i_* and grounding pressure *σ*(*x*), it is also assumed that the track shoes have the same horizontal displacement *s* but different sinkage *z_i_*.

The traction force *T* of the whole crawler equals to the sum of the traction force *T_i_* caused by the horizontal compression and shear of the grouser of every track shoe *i* and expressed by
(8)$$T=\sum_{i=1}^{n}T_{i}$$


In order to calculate the traction force *T_i_* of the grouser, it is assumed that every track shoe *i* has a length *L*, width *B*, grouser height *h*, and a location *x* = *Il-L*/2 in analyzing the kinetic process of a single-track shoe, as shown in Figure 8. When a single-track shoe moves from location I to location II under the action of the horizontal traction force *T_i_* and vertical force 2*BLσ*(*x*), there is a horizontal shear displacement *s* and vertical sinkage *z_i_*. Meanwhile, the horizontal shear stress *τ_i_* changes from *τ_i_*_0_ to *τ_i_*_1_, corresponding to disturbed sediment area *A_i_*_1_ and the vertical compressive stress ranges *σ_i_* from *σ_i_*_0_ to *σ_i_*_1_ corresponding to disturbed sediment area *A_i_*_2_. It can be known that the compressive stress *σ_i_* equals to grounding pressure *σ*(*x*) according to balance of the vertical forces. Hence, the work *W_i_*_1_ done by the shear stress *τ_i_* and the work *W_i_*_2_ done by the compressive stress *σ_i_* can be described by the following two equations, respectively:(9)$$W_{i1}=Bh\int_{0}^{s}\tau_{i}ds$$
(10)$$W_{i2}=BL\int_{0}^{z}\sigma_{i}dz$$


Based on the word-energy principle and Wong’s suggestion [27], it can be known that when a rigid track shoe, width *B* and contact length *h,* moves through a distance *s*, the work *W_i_*_1_′ to make the rut of area *A_i_*_1_′ can be assumed to be equal to the work *W_i_*_2_ necessary to compact the sediment of the area of *A_i_*_2_ corresponding to the contact part of track to the sinkage *z_i_*. Therefore, the relationships between the two works can be given as follows:*W_i_*_2_ = *W_i_*_1_′(11)

Hence, there are two kinds of works in the horizontal direction after the analysis above, and the sum work *W_i_* (*W_i_* = *T_i_**s*) of them equals to *W_i_*_1_ + *W_i_*_1_′ or *W_i_*_1_ + *W_i_*_2_. Eventually, the expression of traction force *T_i_* under the consideration of vertical compressive stress is written as follows:(12)$$T_{i}=\frac{W_{i1}+W_{i2}}{s}=\frac{Bh\int_{0}^{s}\tau_{i}ds+BL\int_{0}^{z}\sigma_{i}dz}{s}$$


### 3.2. Time-Dependent Traction Force T under Different Grounding Pressure Distributions

When the tracked miner walks on complex deep-sea topographies such as deep-sea mountains and sea ditches, different grounding pressure distributions develop under the crawler of the tracked miner. If the grounding pressure *σ_i_* under every track shoe just changes with the horizontal location *x* but not with the time *t*, it is assumed that the types of grounding pressure distribution, relevant with the weight *G* of the tracked miner, can be simplified into four types: (a) uniform distribution with amplitude $\sigma_{i}(x)=G/BL$, (b) linear decrement distribution with amplitude $\sigma_{i}(x)=2Gx/BL^{2}$, (c) linear increment distribution with amplitude $\sigma_{i}(x)=-2Gx/BL^{2}$, and (d) sine distribution with amplitude $\sigma_{i}(x)=(\pi G/2BL)\sin(\pi x/L)$, as shown in Figure 9.

Taking into account the time-dependent parameter *c*(*t*) of shear strength and grounding pressure distribution *σ_i_* (*x*) to Equations (4) and (12), the sum of time-dependent traction force *T_i_* can be obtained as follows:(a)Uniform distribution, $\sigma_{i}(x)=G/BL$
(13)$$\begin{array}{l} T_{uniformed}=\sum_{i=1}^{n}T_{i}=\sum_{i=1}^{n}\frac{Bh\int_{0}^{s}\tau_{i}ds+BL\int_{0}^{z}\sigma_{i}dz}{s}\\ =\left.\left\{Bh\left[c(t)+\frac{G}{BL}\tan\varphi \right]\left(s+\kappa e^{-s/\kappa }\right)+\frac{Gz}{s}\right.\right\}\frac{n^{2}+n}{2} \end{array}$$
(b)Linear decrement distribution, $\sigma_{i}(x)=2Gx/BL^{2}$
(14)$$\begin{array}{l} T_{decrement}=\sum_{i=1}^{n}T_{i}=\sum_{i=1}^{n}\frac{Bh\int_{0}^{s}\tau_{i}ds+BL\int_{0}^{z}\sigma_{i}dz}{s}\\ =\left.\left\{Bh\left[c(t)+\frac{Gn^{2}}{BL}\tan\varphi \right]\left(s+\kappa e^{-s/\kappa }\right)+\frac{n^{2}Gz}{s}\right.\right\} \end{array}$$
(c)Linear increment distribution, $\sigma_{i}(x)=-2Gx/BL^{2}$
(15)$$\begin{array}{l} T_{increment}=\sum_{i=1}^{n}T_{i}=\sum_{i=1}^{n}\frac{Bh\int_{0}^{s}\tau_{i}ds+BL\int_{0}^{z}\sigma_{i}dz}{s}\\ =\left.\left\{Bh\left[c(t)-\frac{Gn^{2}}{BL}\tan\varphi \right]\left(s+\kappa e^{-s/\kappa }\right)-\frac{n^{2}Gz}{s}\right.\right\} \end{array}$$
(d)Sine distribution, $\sigma_{i}(x)=(\pi G/2BL)\sin(\pi x/L)$
(16)$$\begin{array}{l} T_{sine}=\sum_{i=1}^{n}T_{i}=\sum_{i=1}^{n}\frac{Bh\int_{0}^{s}\tau_{i}ds+BL\int_{0}^{z}\sigma_{i}dz}{s}\\ =\left.\left\{Bh\left[c(t)+\frac{\pi G}{2BL}\sin\frac{\pi n^{2}}{2}\tan\varphi \right]\left(s+\kappa e^{-s/\kappa }\right)+\frac{\pi n^{2}LGz}{4s}\sin\frac{\pi n^{2}}{2}\right.\right\} \end{array}$$



## 4. Verification and Analysis of the Traction Force

### 4.1. Verification of the Traction Force Model

In order to verify the traction force model, the traction force experiment of a grouser is conducted on the apparatus designed by our research team as shown in Figure 10 [28]. It can be seen that simulant soil is prepared in the glass tank with a steel frame, and a grouser with width (4 cm) and height (4 cm) is fixed on a truck connecting with a tension sensor. Then, the constant compressive stress (5 kPa) is applied on the grouser (because there cannot be a distributed grounding pressure for only one grouser) with a constant velocity (*v* = 2 cm/s) from the motor controlled by a motor speed controller. When the truck with four pulleys moves along a rail, the traction force varying with time data from the tension sensor can be collected by the data collector and stored in the computer for analyses. Figure 11 shows a rut shaped by the grouser after the experiment [28].

Figure 12 illustrates the two curves of an experimental way [27] and theoretical way for comparison by adopting the parameters listed in Table 3 (not mentioned parameters above). Since the curve from the experimental way is from one track shoe grouser by interacting with the deep-sea simulant sediment, letting *n* = 1 (only one track shoe grouser) in the Equation (13) under the uniformed grounding pressure, the theoretical traction force curve under the same condition as the experimental way is obtained, and it can be seen that the two curves are close and have basically the same trend of change, which proves that the theoretical values of traction force are reliable. The two curves have the close peak values, and it is important for evaluating the traction force varying with time and proves the reliability of the traction force model.

### 4.2. Influence of Time and Track Shoe Number on the Traction Force under Different Grounding Pressure Distributions

#### 4.2.1. Traction Force (*T*)- Time (*t*) Relationships

Figure 13 shows the relationships between traction force *T* and time *t* of the track shoes (*n* = 10). It can be seen that at the beginning of moving (i.e., *t* = 0 s), the instant traction forces are positive under uniformed distribution and linear decrement distribution conditions, but negative under linear increment distribution conditions. After the tracked miner moves (i.e., *t* > 0 s), the traction force changes greatly with different trends. It illustrates that the traction force is greatly influenced by the time. Moreover, during the period between 0 and 4 s, the traction forces decrease with time under uniformed distribution and linear decrement distribution conditions, but basically keep constant under sine distribution and increase with time under linear increment distribution. All the traction forces arrive at zero at about *t* = 4 s. It can be seen that when *t* > 4 s, the *T_increment_* > *T_sine_ > T_uniformed_ > T_decrement_* during the same period. This is because the track shoes near the front of the crawler head are subject to the largest magnitude of the grounding pressure under the linear increment distribution condition, which leads to *c*(*t*) getting the largest in a short time and helps increase the *T_increment_*. For the sine distribution condition, the traction force under the different periodic grounding pressure is basically eliminated, but the sum of the traction force is not zero because of the effect of time on different track shoes with different locations. Under the uniformed and linear decrement distribution conditions, the *T_uniformed_* and *T_decrement_* are always negative between and 4 s and 10 s and signify that the *T_uniformed_* and *T_decrement_* becomes the resistance for the moving tracker miner, which is due to the smaller effect of temporal summation on the time-dependent parameter of the shear strength.

Under the same grounding pressure distribution, the curve of the traction force under the linear increment and sine distribution conditions are increasing during the main analysis period and beneficial for the moving tracked miner. For other conditions, the corresponding curves become negative and resistance for the tracked miner.

#### 4.2.2. Traction Force (*T*)- Number of Track Shoe (*n*) Relationships

Figure 14 illustrates the relationships between the traction force *T* and the number of track shoe *n* under the different grounding pressure distributions. It can be seen that within the 10 s, the magnitudes of *T* are increasing or decreasing obviously with *n,* except the uniformed condition. The cause is that the increase of *n* leads to more grounding pressure applied on the track shoes and develops more work or negative work to overcome the corresponding sinkage; for the uniformed increment condition, less contact time with sediment of larger grounding pressure weakens the magnitude and change of *T* and leads to slight traction force change.

Moreover, under the same number of track shoe *n* and during the same period of time *t*, the absolute value relationships of *T* are *T_sine_ > T_increment_* > *T_decrement_* > *T_uniform_*. For the sine condition, the curve of *T* is close to sine and gradually increases, but with an abrupt decrease as the number of track shoe becomes 10. This is because the increasing *n* in the traction force model may narrow the sine characteristics based on the fact that the maximum value of sine is one.

## 5. Conclusions


(1)A semi-empirical time-dependent shear strength (cohesion force) of describing the shear strength can be obtained by letting the compressive stress be zero in direct shear rheological constitutive model based on the analysis of Mohr–Coulomb shear strength theory and direct shear rheological experiment. Several time-dependent traction force models under the different grounding pressure distributions are deduced with the time-dependent cohesion force based on the work-energy principle. The models take the time, grounding pressure, and track shoe number into account and is used for conveniently analyzing the influence of kinds of key parameters on traction force of the deep-sea tracked miner.(2)The traction force model is verified by a comparison between an experimental curve and a theoretical curve of a single-track shoe. By analyzing the influence of time and track shoe number on time-dependent traction force, it is found that *T_increment_* > *T_sine_ > T_uniformed_* > *T_decrement_* when *t* > 4 s under different grounding pressure distributions. The linear increment grounding pressure distribution is suggested for evaluating the most favorable traction force and the linear decrement grounding pressure distribution for calculating the worst traction force. Both grounding pressure distributions can better help the crawler design and optimization for better trafficability and stability of the deeps-sea tracked miner when adopting the time-dependent cohesion force.(3)The traction force calculation is proved to be valid by the traction force experiment of a single-track shoe, and the influence of time, number of the track shoe, and grounding pressure distribution on the traction force can provide scientific basis for designing the crawler and evaluating the trafficability of tracked miner.


## Figures and Tables

**Figure 1 sensors-22-01119-f001:**
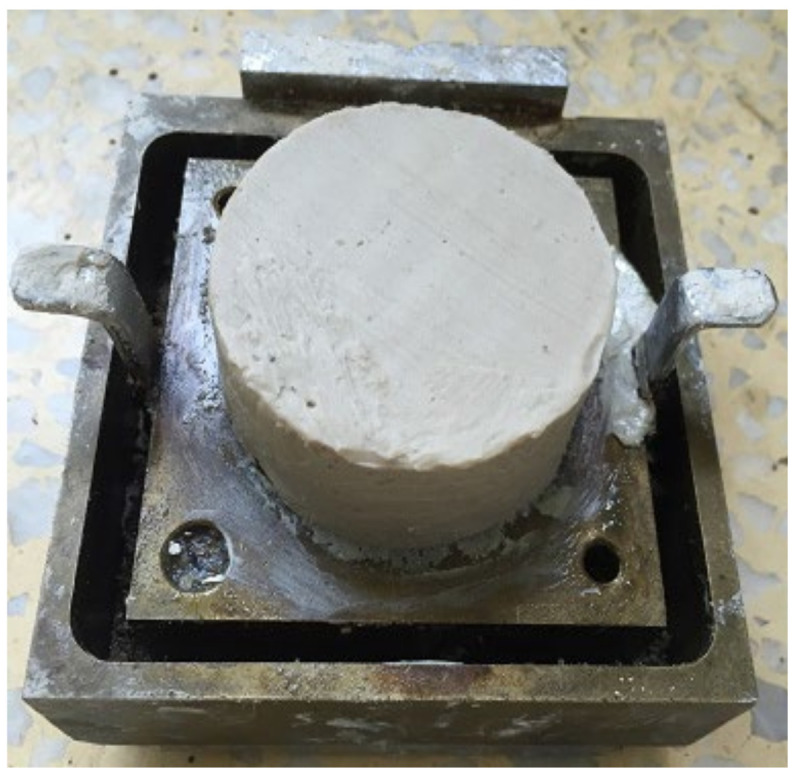
Standard cylinder sample of deep-sea simulant sediment.

**Figure 2 sensors-22-01119-f002:**
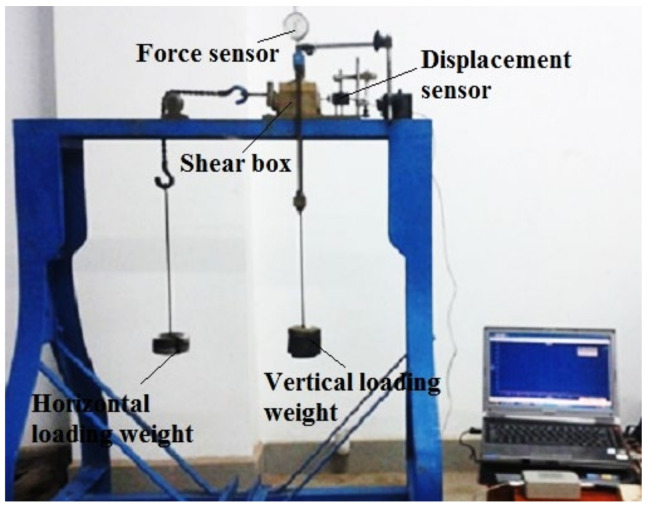
Direct shear creep apparatus of deep-sea simulant sediment.

**Figure 3 sensors-22-01119-f003:**
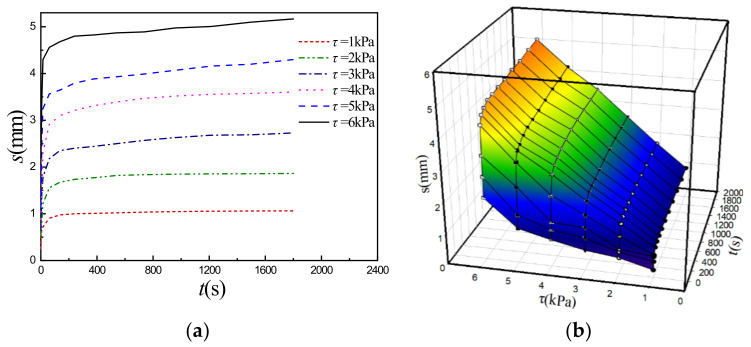
Shear creep curves and fitted 3D surface at *σ* = 5 kPa. (**a**) Shear creep curves under *τ*; (**b**) Fitted curved 3D surface.

**Figure 4 sensors-22-01119-f004:**
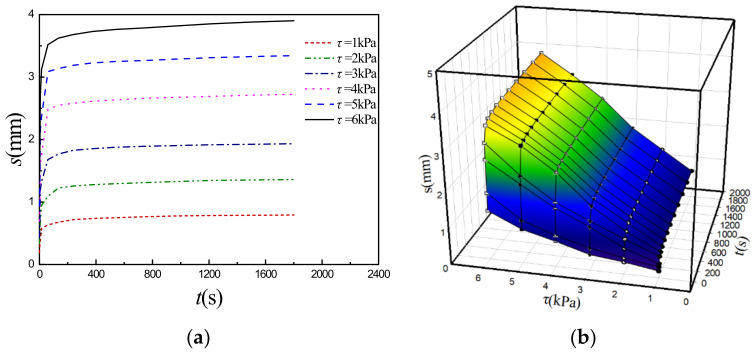
Shear creep curves and fitted 3D surface at *σ* = 10 kPa. (**a**) Shear creep curves under *τ*; (**b**) Fitted curved 3D surface.

**Figure 5 sensors-22-01119-f005:**
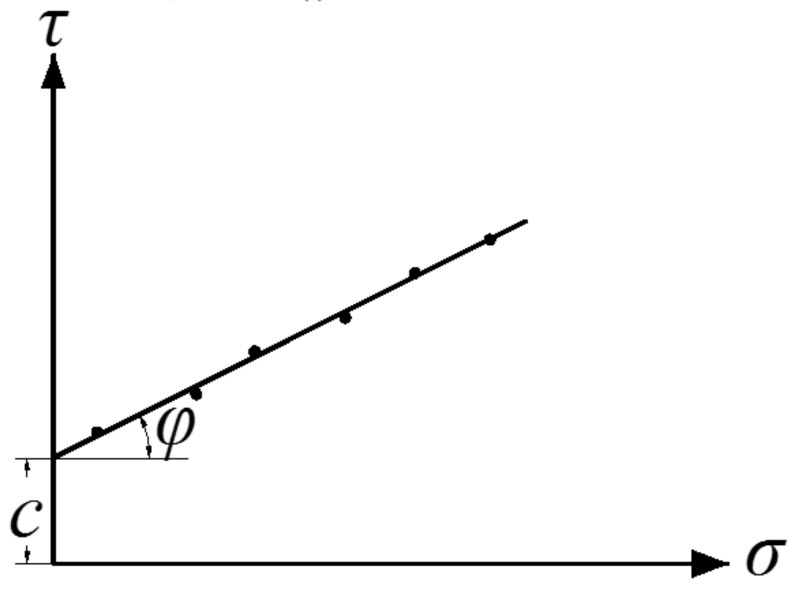
Determination of the cohesion force c and friction angle *φ*.

**Figure 6 sensors-22-01119-f006:**
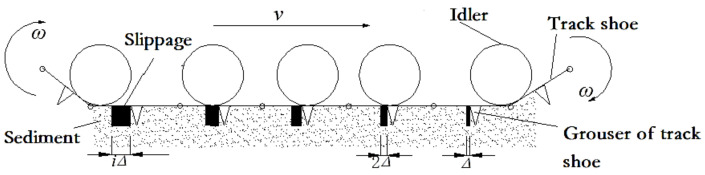
Simplified model of tracked miner’s motion.

**Figure 7 sensors-22-01119-f007:**
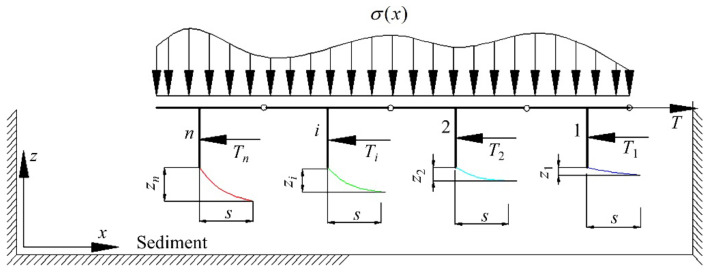
Simplified mechanic model of the crawler.

**Figure 8 sensors-22-01119-f008:**
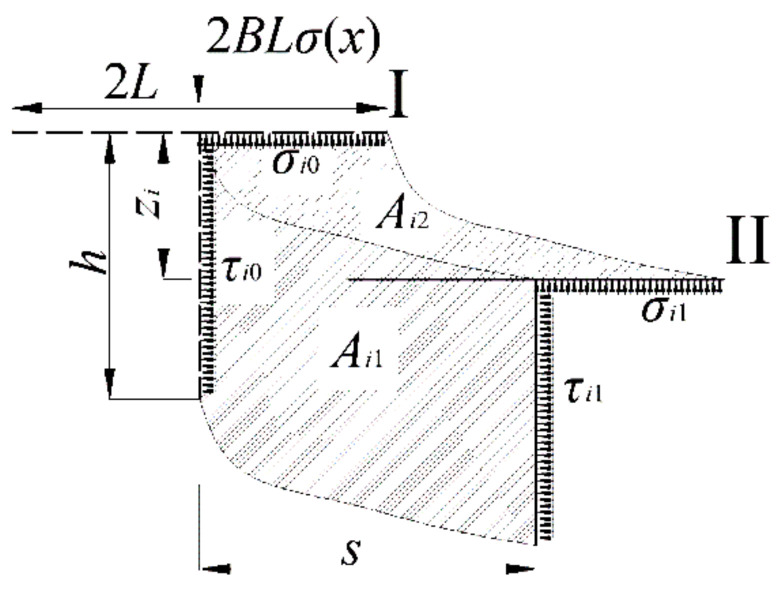
Kinetic process of single-track shoe.

**Figure 9 sensors-22-01119-f009:**
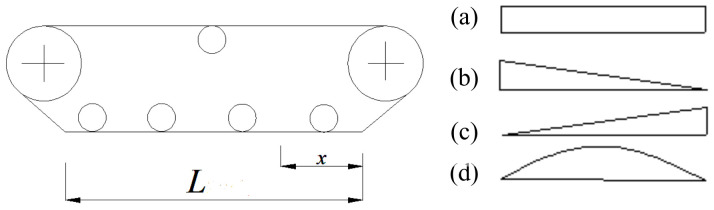
Four kinds of grounding pressure distribution under the crawler with length *L* [13].

**Figure 10 sensors-22-01119-f010:**
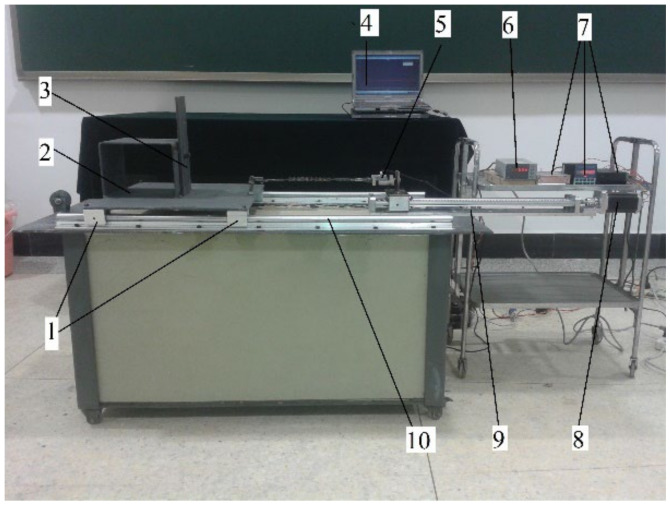
Traction force experiment apparatus of track shoe grouser. 1, Pulley; 2, Truck; 3, Grouser; 4, Computer; 5, NS-WL1 tension sensor; 6, Data collector; 7, Motor speed controller; 8, Motor; 9, Screw rod; 10, Rail.

**Figure 11 sensors-22-01119-f011:**
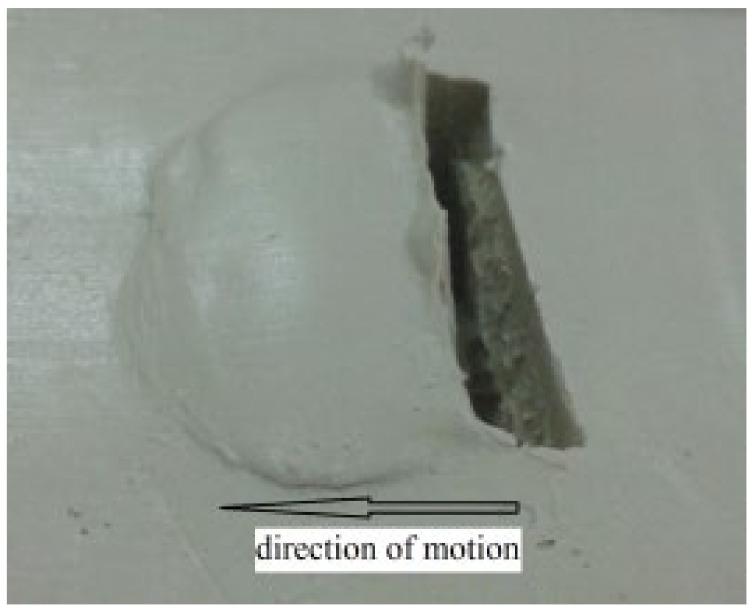
A rut shaped by the grouser (the grouser is removed).

**Figure 12 sensors-22-01119-f012:**
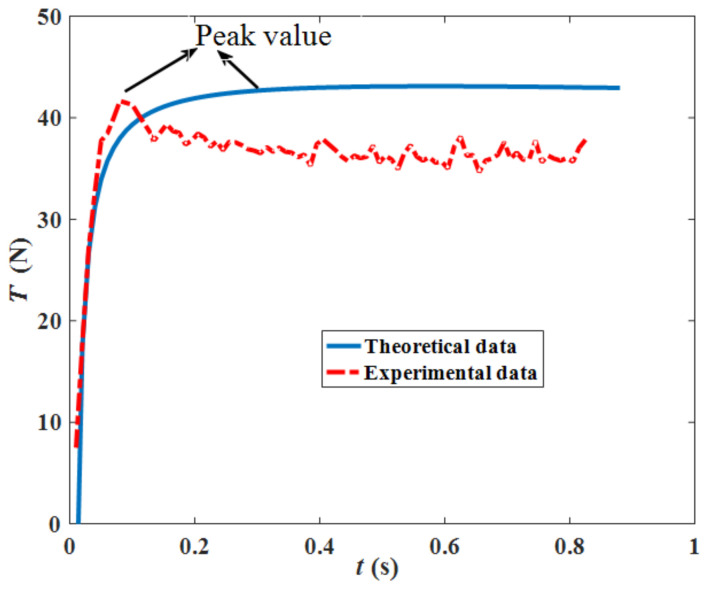
Comparison of the theoretical data and experimental data.

**Figure 13 sensors-22-01119-f013:**
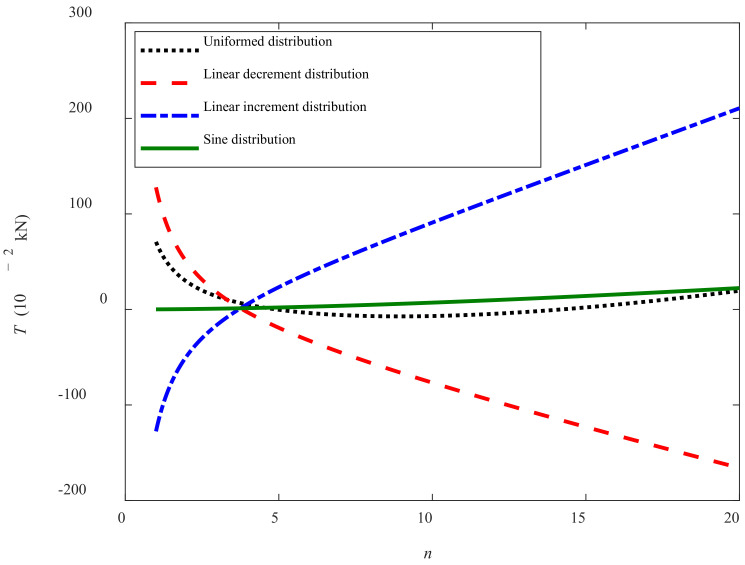
Curves of traction force vs. time.

**Figure 14 sensors-22-01119-f014:**
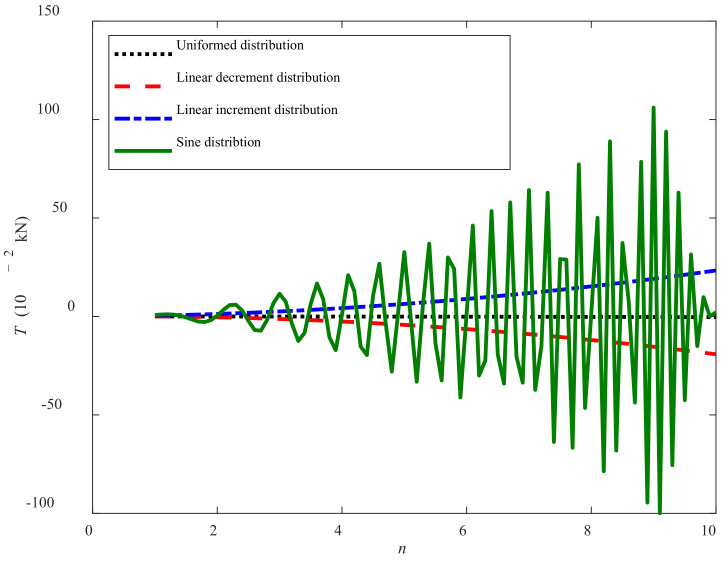
Curves of number of track shoe vs. traction force.

**Table 1 sensors-22-01119-t001:** Main physical and mechanical parameters of sediment.

Physical and Mechanical Parameters	Simulant Sediment	Undisturbed Sediment
Wet density, *ρ*/(t·m^−3^)	1.315	1.250
Water content, *w/*%	165.6	246.5
Liquid limit, *w_L_*/%	190.2	145.2
Cohesion force, *c*/(kPa)	6.2	6.0
Friction angle, *φ*/(°)	1.72	3.1
Penetration resistance, *P_s_*/(kPa)	87	50–90

**Table 2 sensors-22-01119-t002:** Fitted direct shear creep parameters.

*σ*/kPa	$K_{1}(\sigma )/\mathrm{MPa}$	$K_{2}(\sigma )/\mathrm{MPa}$	$\beta_{1}(\sigma )/(\mathrm{MPa}\cdot s)\;\times \;10^{3}$	$\beta_{2}(\sigma )/(\mathrm{MPa}\cdot s)\;\times \;10^{3}$	R-Square
5	7.36	1.82	7.38	22.90	0.987
10	10.27	2.25	10.24	27.50	0.992
15	11.20	2.95	11.26	24.73	0.984
20	11.22	5.51	11.41	304.04	0.995
25	13.29	8.09	13.90	421.70	0.984
30	19.24	9.87	19.15	466.84	0.983

**Table 3 sensors-22-01119-t003:** Main size of crawler in time-dependent traction force model.

Length *L* (m)	Width *B* (m)	Weight *G* (kN)
6	1.7	110

## Data Availability

All data included in this study are available upon request by contact with the first author or corresponding author.

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
