# Peer review of "Semi-Empirical Time-Dependent Parameter of Shear Strength for Traction Force between Deep-Sea Sediment and Tracked Miner"

_sensors, 2022, doi:10.3390/s22031119_

Round 1

Reviewer 1 Report

The presented work conducts a study on direct shear creeps to obtain the shear strength parameters. The scientific level of the work is good, however, the novelty has not been expressed clearly. 

  • Your article would appear to be of interest to a wide engineering research community and in order to promote its visibility, even more, I recommend that you view the past published articles in Sensors Journal and if you find any relevant publications, CITE the article from this Journal.
  • There are some mistakes in spelling and grammar. The English language of the paper should be checked and revised by a native speaker.
  • line 133, add the reference instead of writing: (Xu et al., 2018a).
  • The quality of Fig.2 is very low. please consider replacing it. The same for Fig.10.
  • The organization of the results/conclusions section should be improved.  The discussion section is not clear when discussing data. Please consider rewriting this section.

Author Response

Response to Reviewer 1 Comments:

Comment 1: I recommend that you view the past published articles in Sensors Journal and if you find any relevant publications, CITE the article from this Journal.

Response: Thank you for your suggestions, we have cited an article in reference 1.

Comment 2: There are some mistakes in spelling and grammar. The English language of the paper should be checked and revised by a native speaker.

Response: According to the reviewer’ s comment, we have invited a native speaker to checked and revised our spelling mistakes.

Comment 3: line 133, add the reference instead of writing: (Xu et al., 2018a).

Response: According to the reviewer’ s suggestion, we have added the reference instead of writing: (Xu et al., 2018a) and marked it in red.

Comment 4: The quality of Fig.2 is very low. please consider replacing it. The same for Fig.10.

Response: According to the reviewer’ s suggestion, we have adopted new figures with enhanced quality.

Comment 5: The organization of the results/conclusions section should be improved. The discussion section is not clear when discussing data. Please consider rewriting this section.

Response: According to the reviewer’ s suggestion, I have rewritten the results/conclusions section and marked them in red.

Reviewer 2 Report

In this paper, experimental studies of direct shear creep were carried out to obtain semi-empirical time-dependent shear strength parameters based on the analysis of the rheological direct shear model and the Mohr-Coulomb theory of shear strength.

The research presented in the article is of utilitarian importance when designing the crawler and evaluating the trafficability of tracked miner.

The introduction sufficiently describes the current state of knowledge with reference to References and justifies taking up the research topic.

Keywords: I believe that keywords are too general and should characterize the content of the article in more detail.

Chapter 2.1, lines 119-133: The description is incomprehensible:

1) No reference in the text to Fig. 2.

2) It is not understood what the meanings are: “(part a) (part e,) (part e ',), (component b and b'), (part c and c ') (part d and d'). (part f) "

3) “(Xu et al., 2018a)” - the reference to literature is incorrectly spelled.

Fig. 10: The details of the device structure are difficult to identify in the photo. I believe that an interesting part of the device should be enlarged so that the elements of its construction are visible or a diagram should be included.

Author Response

Response to Reviewer 2 Comments

Comment 1: Keywords: I believe that keywords are too general and should characterize the content of the article in more detail.

Response: According to the reviewer’ s suggestion, I have modified part of the keywords and make it more appropriate.

Comment 2: Chapter 2.1, lines 119-133: The description is incomprehensible:

1) No reference in the text to Fig. 2.

2) It is not understood what the meanings are: “(part a) (part e,) (part e ',), (component b and b'), (part c and c ') (part d and d'). (part f) "

3) “(Xu et al., 2018a)” - the reference to literature is incorrectly spelled.

Fig. 10: The details of the device structure are difficult to identify in the photo. I believe that an interesting part of the device should be enlarged so that the elements of its construction are visible or a diagram should be included.

Response: Thanks for your valuable advice and according to the reviewer’ s suggestions,

1) I added a reference in the text marked it in red (line 121);

2) I deleted the redundant “(part a) (part e,) (part e ',), (component b and b'), (part c and c ') (part d and d'). (part f) ".

3)I modified the reference to literature “(Xu et al., 2018a)” with “[7]”. As for the device structure, there are some difficulty in finding an enlarged version to make it visible, but I have rewritten the relevant sentences to make the device structure clearly.

Reviewer 3 Report

A file has been uploaded with suggestions

Author Response

Response to Reviewer 3 Comments

Comment 1: The article presents a proprietary method of determining the distance for the problem of finding a miner in a tunnel. I have no substantive comments as to the whole article, however, the formula No. 6 is incomprehensible. Near the symbol "c" there is an artifact which, in my opinion, should be removed or replaced with a mathematical symbol.

Response: Thanks for your valuable advice. In the formula No. 6, I replaced symbol "c" with “c(t)”.

Reviewer 4 Report

The article presents a proprietary method of determining the distance for the problem of finding a miner in a tunnel. I have no substantive comments as to the whole article, however, the formula No. 6 is incomprehensible. Near the symbol "c" there is an artifact which, in my opinion, should be removed or replaced with a mathematical symbol. 

Author Response

Response to Reviewer 4 Comments

Comment 1: A thorough text review is mandatory before publication. Some explanations are really difficult to follow. Particularly, the core model description (Eqs. 13 to 16) are not at all explained. This comment is in direct relationship to the following one.

Response: Thanks for your valuable advice. Since taking into account time-dependent parameter c(x) of shear strength and grounding pressure distribution  i (x) to Eq. (4) and Eq. (12), Eq. 12 is the core model and it is deduced from fact below:  Based on the word-energy principle and Wong’s suggestion [26], when a rigid track shoe width B and contact length h moves through a distance s, the work Wi1′to make the rut of area Ai1′can be assumed to be equal to the work Wi2 necessary to compact the sediment of the area of Ai2 corresponding to the contact part of track to the sinkage zi. (line 235 to line 244)

Comment 2: A symbol list would also be a must in all Journal articles.

Response: Thanks for your valuable advice. I have followed the requirement of the journal.

Comment 3: The sentences in line 9, similarly to sentence in line 96 and in the first conclusion (line 347) include three times the word “shear”. Probably a substitute or equivalent word could be used to reduce the number of “shear” in the same sentence.

Response: Thanks for your valuable advice. The “shear” is a key word and I am afraid it is difficult to find a substitute or equivalent word.

Comment 4: There is a word “Wight” in Table 3.

Response: The wrong word “Wight” is modified into “weight”

Comment 5: Figure 12 is not well explained (I think). The theoretical values, do they come from the equations (13) to (16)? How are the experimental values obtained? What is the possible comment on the pointed as peak value?

Response: I Have rewritten the explanation of Figure 12 (line 290-line 299). The theoretical values just come from the equations (13) and I have made it clear in the explanation. The experimental values is obtained by way involved in the reference [27]. The peak value can be influenced by many factors where the property of the sediment is predominating in deciding its peak value based on our previous study. 

Comment 6: Probably on table 1, the units are better understood using parenthesis, for instance: Cohesion force, c (kPa).

Response: Thanks for your valuable advice. I used parenthesis on table 1.

Comment 7: On the lay-out of the values, for instance (line 139, but there are many more)=5 kPa it would be better to include some spacings as: =5kPa.Also, in thesame page, lines 151 and 152 are 1 and 2, that should be 1 and 2.

Response: Thanks for your valuable advice. I have included some spacings in the corresponding lines.

Comment 8: A lack of uniformity is found in the references. Some of the are with initial capital letters (#2 and #5, for instance), while others are in low letters. Reference #26 lacks some details on pages, etc.

Response: Thanks for your valuable advice. I have made the references uniformed and add some details in the Reference #26.

Round 2

Reviewer 1 Report

I see that some modifications have been performed, which makes this paper qualified to be published in the Journal after some English spell-check.